# Bridging Information-Theoretic and Geometric Compression in Language Models

**Emily Cheng** and **Corentin Kervadec**
Universitat Pompeu Fabra / Barcelona
{name.lastname}@upf.edu

**Marco Baroni**
ICREA / Barcelona
marco.baroni@upf.edu

## Abstract

For a language model (LM) to faithfully model human language, it must compress vast, potentially infinite information into relatively few dimensions. We propose analyzing compression in (pre-trained) LMs from two points of view: geometric and information-theoretic. We demonstrate that the two views are highly correlated, such that the intrinsic geometric dimension of linguistic data predicts their coding length under the LM. We then show that, in turn, high compression of a linguistic dataset predicts rapid adaptation to that dataset, confirming that being able to compress linguistic information is an important part of successful LM performance. As a practical byproduct of our analysis, we evaluate a battery of intrinsic dimension estimators for the first time on linguistic data, showing that only some encapsulate the relationship between information-theoretic compression, geometric compression, and ease-of-adaptation.

## 1 Introduction

To speak a language is not to memorize all possible utterances, but to instead extract the finite ruleset and lexicon that generates them (Chomsky, 1986). That is, language, though nominally high-dimensional, can be compressed to a comparatively small *intrinsic dimension*.

The recent success of *(large) language models* demonstrates that artificial neural networks, too, can acquire linguistic knowledge. Current language models (LMs) are Transformer-based architectures at-scale (Vaswani et al., 2017) that are trained on the conditional distribution of natural language (OpenAI, 2023; Zhang et al., 2022; Touvron et al., 2023; Chowdhery et al., 2022) and that are inching closer to human-like linguistic robustness (Brown et al., 2020; Liang et al., 2022; Wang et al., 2019b). For an LM to faithfully model language, it must encode linguistic training data into finitely many variables that allow generalization to infinitely many

grammatical utterances. That is, it must perform a successful form of data compression.

Thus, following the line of research that aims to better understand LM behavior (Wei et al., 2022; Zhang et al., 2021; Rogers et al., 2020), in this work we provide initial insights on how they *compress linguistic knowledge*. We demonstrate an empirical link between two types of compression: geometric and information-theoretic. In particular, we ask: how, and by how much, do LMs compress linguistic data? Furthermore, what are linguistic correlates to compressibility? Is compression a good predictor of rapid adaptation? We show that (1) intrinsic dimension (ID) of linguistic data representations under an LM tracks information-theoretic coding length; (2) greater data compression predicts ease-of-adaptation in causal language modeling tasks; (3) interpretable linguistic properties such as vocabulary size and syntactic structure modulate ID; and (4) different model sizes recover similar ranges of ID. Finally, as a practical contribution, (5) we explore different ways to estimate ID of linguistic data, and find only some to capture the relation between ID, coding length, and ease-of-adaptation.

## 2 Related Work

**Causal Language Models** State-of-the-art language models are based on the Transformer architecture (Vaswani et al., 2017), which consists of alternating feed-forward and self-attention modules (Bahdanau et al., 2015). They are trained using *self-supervised learning* on sequences of *tokens*, where a token is defined as the atomic unit (e.g., a word or a sub-word) fed into the language model. Due to their current ubiquity, we focus on autoregressive models trained on a *causal language modeling* objective, that is, next token prediction given a context of previous tokens (Brown et al., 2020; Radford and Narasimhan, 2018; Zhang et al., 2022).

Language models are typically measured against human performance on linguistic benchmarks.

Evaluation may be done post-finetuning or in a low-shot regime, where the model completes a linguistic task given few or zero examples. A variety of benchmarks, such as GLUE (Wang et al., 2019b), SuperGLUE (Wang et al., 2019a), and BigBENCH (Srivastava et al., 2022) have been proposed, evaluating, for instance, model performance on textual entailment, question-answering, semantic equivalence, or sentiment analysis. Indeed, human baselines for GLUE and SuperGLUE have already been surpassed by LMs that contain billions of parameters (e.g., PaLM 540B (Chowdhery et al., 2022)).

## 2.1 Compression in LMs

There is a wide body of work analyzing compression in deep neural architectures. In statistical learning theory, compression has been empirically and theoretically linked to generalization (Shwartz-Ziv and Tishby, 2017; Arora et al., 2018). Moreover, deep learning models are thought to minimize description length (Perez et al., 2021; Voita and Titov, 2020; Blier and Ollivier, 2018). In large LMs, implicit compression of neural network parameters has been linked to ease-of-finetuning and generalization (Aghajanyan et al., 2021). Our work complements this line of research by focusing on compression of *data representations* rather than network parameters. We consider compression from two different perspectives: information-theoretic and geometric (intrinsic dimension).

**Information-theoretic compression** Compression in neural networks can be quantified from an information-theoretic point of view (Shannon, 1948). For instance, the *information plane* (IP), a widely studied framework introduced by Shwartz-Ziv and Tishby (2017) and Tishby and Zaslavsky (2015), quantifies compression per-layer as the mutual information between representations and inputs. However, there is little consensus in the relevant literature on the appropriate estimator to measure this *internal* compression of inputs (Saxe et al., 2018; Goldfeld et al., 2019; Noshad et al., 2019; Chelombiev et al., 2019; Geiger, 2022).

Instead, as probabilistic models, LMs are natural *black-box* compressors: the negative log-likelihood of the next token given context is, by definition, its Shannon coding length in bits (Shannon, 1948). Concurrent work explores the equivalence between self-supervised prediction and lossless compression, demonstrating that LMs can be powerful general-purpose compressors (Delétang et al.,

2023). We similarly focus on *information-theoretic coding length* of inputs under a pre-trained model, which is a simple measure of compression that, to our knowledge, has not been shown to be analytically equivalent to geometric compression. We develop this measure further in section 3.2.

**Intrinsic Dimension (ID)** *Geometric* compression is commonly quantified using dimensionality reduction techniques. Often underlying these approaches is the manifold learning hypothesis (Goodfellow et al., 2016), or the notion that real-life, high-dimensional data often lie on a low-dimensional manifold. Intrinsic dimension (ID), or the number of degrees of freedom in the data, is the dimension of this data manifold.

Perhaps the most prototypical ID estimators are linear projective methods like random projection (Li et al., 2018) or Principal Component Analysis (PCA) (Jolliffe, 1986). While these project data to a *linear* subspace, the underlying geometric object need not be linear; therefore, e.g., PCA poorly estimates ID for curved manifolds (Campadelli et al., 2015). Nonlinear ID estimators include Correlation Dimension (Grassberger and Procaccia, 1983); Fisher Separability (Albergante et al., 2019); and a host of "nearest-neighbor" (NN)-based methods, which use the fact that manifolds look locally Euclidean to fit the ID based on local neighbor distributions (Facco et al., 2017; Levina and Bickel, 2004; Haro et al., 2008; Amsaleg et al., 2018). Such methods outperform linear ones on ID estimation benchmarks (Campadelli et al., 2015). In section 4, we will assess these methods in the context of linguistic data, and analyze how each of them relates to coding length and ease-of-adaptation.

In deep learning, there has been recent interest in using ID to characterize learning complexity. Intrinsic dimension has been quantified for neural network parameters (Li et al., 2018), as well as for input data and their representations in visual and protein-sequence domains (Cohen et al., 2020; Recanatesi et al., 2019; Ansuini et al., 2019; Valeriani et al., 2023; Pope et al., 2021). These studies show that deep neural architectures learn low-dimensional structures, encoding parameter weights and training data into orders-of-magnitude lower ID than their ambient dimension.

In the linguistic domain, low ID of LM *parameters* has been shown to underlie efficient task adaptation (Aghajanyan et al., 2021), where optimization occurs in low-dimensional, task-specific sub-

spaces (Zhang et al., 2023). Moreover, parameter redundancy in pre-trained LMs can be exploited to design parameter-efficient finetuning methods such as LoRA (Hu et al., 2022). We are interested in ID of *data representations* as opposed to *LM parameters*, as (1) we want to study how different linguistic properties affect their coding; (2) ID estimation of model parameters can be expensive– large LMs can have billions of parameters, while input representations are lower-dimensional, e.g., $D = 4096$ in OPT-6.7b (Zhang et al., 2022). In related work on LM *representation* ID, contextual word embeddings have been found to lie in low-dimensional linear subspaces (Mamou et al., 2020; Hernandez and Andreas, 2021). Most similar to our work, Cai et al., 2021 show that Transformer embeddings of the Wikitext and Penn TreeBank datasets constitute nonlinear manifolds of ID $\sim \mathcal{O}(10)$.

## 3 Methods

Our work attempts to bridge notions of geometric and information-theoretic compression of linguistic data under an LM, and subsequently relate these to ease-of-adaptation. We do so by quantifying the ID of data representations, information-theoretic coding length of linguistic inputs under an LM, and ease-of-finetuning in order to determine whether these three phenomena are correlated.

**Notation** Let a linguistic dataset $X = \{x^{(i)}\}_{i=1}^{N}$ consist of $N$ sequences of tokens, where each sequence $x^{(i)}$ has length $l(x^{(i)})$. Let $\mathcal{M}$ be a (pre-trained) causal language model described by $p_{\mathcal{M}}(\cdot|x_{<j})$, the conditional probability distribution of the $j^{\text{th}}$ token given its past context of tokens.

**Models & Datasets** Experiments are performed for the product of models $\mathcal{M} \in$ [OPT-350m, OPT-1.3b, OPT-6.7b] and datasets $X \in$ table 1. We focus on OPT suite of causal language models due to their accessibility (Zhang et al., 2022). For the datasets, we start from a list of corpora including the GLUE and SuperGLUE benchmarks, then pick those whose size is large enough for ID estimates to converge ($N \geq 10000$). Then, for computational efficiency, we randomly subsample each dataset to size $N' = \max(N, 50000)$, where 50000 is chosen conservatively based on preliminary analyses of convergence of bootstrapped ID estimates (see appendix A.2).

In addition to external datasets, we create one baseline dataset per model, which we call OPTCor-

| Benchmark | Datasets |
|---|---|
| GLUE | cola, mnli, mrpc, qnli qqp, rte, sst2, stsb |
| SuperGLUE | boolq, multirc, wic |
| | IMDB, Penn Treebank, Bookcorpus, Wikitext fr, Tweets, Pile-10k, CNN Dailymail, Openwebtext-10k, CONCODE, OPTCorpus, OPTCorpus-permuted, OPTCorpus-swapped, OPTCorpus-random, Wikitext, Wikitext-permuted, Wikitext-swapped, Wikitext-random |

Table 1: List of datasets used in experiments. We use all datasets for ID and PPL estimation and the ones in the last block for finetuning (except for OPTCorpus, which already reflects the distribution of the model).

pus, of $\sim 24$ million tokens by repeatedly randomly sampling from $\mathcal{M}$ until [EOS] is reached. The conditional next-token distribution of OPTCorpus approximates that of $\mathcal{M}$ so to serve as a reference datapoint.

In order to determine the effects of syntax and lexical semantics on compression, we define three transformations which we apply to a *dataset*: (1) *dataset-permuted:* for each sequence in *dataset*, randomly permute its tokens. This ablates syntax to retain bag-of-tokens lexical information. (2) *dataset-swapped*: excluding special tokens, create a random permutation $\sigma$ over the vocabulary. For each sequence in *dataset*, deterministically map each token by $\sigma$. This ablates lexical patterns, retaining syntactic structure. (3) *dataset-random*: randomly replace each token in *dataset* with another (excluding special tokens). This ablates both syntactic and lexical structure.

Notably, several shallow linguistic descriptors are preserved with these transformations: dataset size, sequence length and vocabulary size (1-3), vocabulary entropy (1,2), and token frequency (1). We apply the transformations to OPTCorpus and wikitext, producing six additional datasets.

### 3.1 Intrinsic Dimension

Given model $\mathcal{M}$ and dataset $X$, we estimate the representational ID of $X$ under $\mathcal{M}$ as follows (see also fig. 1):

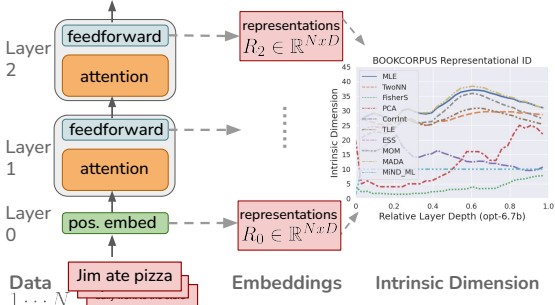

Figure 1: ID estimation. Data (bottom left) are fed into an LM with $M$ blocks (left). Activations post-embedding/[feedforward, attention] blocks $i = 1 \cdots M$ are extracted and aggregated across the sequence length to produce an $N \times D$ dimensional matrix of representations $R_i$. Then, the ID of each $R_i$ is estimated to produce an ID "profile" (right plot).

1. **Preprocess data.** Let the context window length of $\mathcal{M}$ be $l_{\mathcal{M}}$. We preprocess $X$ by splitting all $x^{(i)}$ with length $l^{(i)} > l_{\mathcal{M}}$ into sequences with maximum length $l_{\mathcal{M}}$.

2. **Gather representations.** Evaluate $\mathcal{M}(X)$, gathering intermediate representations after contextualized embedding and after each attention+feed-forward block. In particular, representations are extracted after the residual connection and LayerNorm.

3. **Aggregate representations.** Because input sequences $x^{(i)}$ are variable-length, we use the vector associated to the last token of each layer to represent it. Due to auto-regressive self-attention, the last token in the sequence is the only one to incorporate information from all other tokens in the sequence. Moreover, in the context of causal language modeling, the last token representation is the one used for next-token prediction in the top layer of the model, so it is the one where all information relevant to the prediction task should be concentrated. We leave testing alternative aggregation strategies, such as average pooling (cf. Valeriani et al., 2023) to future work.

   After the aggregation step, we have dataset representations

   $$\mathbf{R} := \{R_j\}_{j=1}^M; R_j \in \mathbb{R}^{N \times D},$$

   where $D$, the hidden dimension of the model, is the ambient (extrinsic) dimension of data representations, and $\mathcal{M}$ has $M$ layers.

4. **Estimate ID.** Per layer $j$, compute the ID

$d_j$ of $R_j$ using ID estimator $g : \mathbb{R}^{N \times D} \rightarrow \mathbb{Z}_+; R_j \mapsto d_j$.

We test 12 different ID estimators $g$, grouping them into categories based on technique: nine NN-based (Facco et al., 2017; Farahmand et al., 2007; Carter et al., 2010; Amsaleg et al., 2019, 2018; Haro et al., 2008; Johnsson et al., 2015; Rozza et al., 2012; Ceruti et al., 2014), one projective (PCA), one based on fine-grained clustering (Fisher Separability, Albergante et al., 2019), and one fractal-based (Correlation Dimension, Grassberger and Procaccia, 1983). Further details on estimators can be found in appendix A.1. We implement all estimators using the skdim Python package (Bac, 2020).

### 3.2 Information-Theoretic Compression

Information-theoretic compression is directly related to the training objective of the model, which minimizes the average negative log-likelihood loss of next-token prediction over the training set.

**Learning minimizes coding length** The average negative log-likelihood (NLL) training objective of causal LMs is given by

$$\min_\theta \frac{1}{\sum_{i=1}^N l(x^{(i)})} \sum_{i=1}^N \sum_{j=1}^{l(x^{(i)})} - \log p_{\mathcal{M}}(x_j^{(i)} | x_{<j}^{(i)}; \theta), \tag{1}$$

that is, to minimize the empirical negative log-likelihood of the next token given its context with respect to model parameters $\theta$. This is analytically equivalent to minimizing the average number of bits to encode the $j^{\text{th}}$ token under $p_{\mathcal{M}}$.

We are interested in quantifying information-theoretic compression at the sequence level. We do so by using perplexity, a common metric in NLP.

**Perplexity** The perplexity (PPL) of a sequence $x^{(i)}$ is the exponentiated negative log-likelihood loss

$$PPL^{(i)} := 2^{\frac{1}{l(x^{(i)})} \sum_{j=1}^{l(x^{(i)})} - \log p_{\mathcal{M}}(x_j^{(i)} | x_{<j}^{(i)}; \theta)}. \tag{2}$$

We compute the average PPL for each dataset $X$ by performing forward passes through $\mathcal{M}$. As PPL is monotonic in coding length, we use PPL as our measure of interest to proxy information-theoretic compression, later relating this quantity to the representational ID of $X$.

### 3.3 Ease-of-Adaptation

Low ID of pre-trained LM parameters has been shown to predict ease-of-finetuning (Aghajanyan et al., 2021). We complement this finding by correlating the ID of *data representations* to an LM's ease-of-adaptation to that dataset.

Ease-of-adaptation to a downstream task depends not only on the inputs $X$ but also on task-specific outputs. For instance, binary classification can be less complex than causal language modeling given the same inputs $X$. As it is not always clear what is the best way to encode the outputs in order to measure the quantities of our interest, we focus on adaptation under a causal language modeling objective, which is the same as the model's pre-training objective. This entails little loss of generality, as task adaptation is nowadays commonly framed as a language model adaptation problem.

**Adaptation procedure** We perform finetuning for each of OPT-350m, 1.3b, and 6.7b on the datasets $X$ in table 1 that are suited to causal language modeling, i.e., omitting [Super]GLUE.

Due to resource constraints, and as we compare between datasets and not models, we perform full finetuning for OPT-350m and finetune using LoRA (Hu et al., 2022) for the larger sizes. We end finetuning at a maximum of 15 epochs or when validation loss converges. Loss is considered to have converged as soon as it fails to decrease for 3 evaluation steps, each 500 iterations apart. Detailed hyperparameter settings may be found in appendix C.

**Adaptation metrics** We quantify ease-of-adaptation with the following, where $T$ is defined as the number of iterations until convergence:

1. $PPL_T$: final evaluation perplexity.
2. Sample complexity $S = \frac{1}{T}\sum_{t=1}^{T} PPL_t$, where $PPL_t$ is the evaluation PPL at evaluation step $t$.

Finally, we compute Spearman correlations between these metrics, zero-shot perplexity ($PPL_0$), and ID to assess whether the two types of compression and ease-of-adaptation are linked.

## 4 Results

We find that, similar to protein models and visual networks (Ansuini et al., 2019; Valeriani et al., 2023), the ID of linguistic data representations

is significantly lower than their ambient dimension: in our case, by roughly 2 orders of magnitude. In particular, while the ambient dimension of representations in OPT-350m, OPT-1.3b, and OPT-6.7b are $D = 1024, 2048$, and 4096, respectively (Zhang et al., 2022), dataset representational ID is $d = \mathcal{O}(10)$, see fig. 5.

For simplicity, in the main article we present results on one representative NN-based estimator, the Expected Simplex Skewness (ESS) method of Johnsson et al., 2015, as it is the only one to significantly correlate with all other estimators ($\alpha = 0.1$) (see fig. 6).[1] We present results on other estimators in appendix E, and we comment on other (non-NN-based) ID estimators in section 4.4. Also for practicality, we report in the main article results obtained with one model, OPT-6.7b, only commenting on other models when relevant, and one aggregated ID measure across layers: the *max* ID value, seen as a conservative upper bound on ID across layers. This choice is supported by the observation that, for most datasets, the ID profile over layers is quite flat (appendix D). Results with other aggregated measures are presented in appendix E.

Our primary result is that, in the pre-trained LMs tested, information-theoretic compression (PPL) predicts geometric compression (ID), and low ID predicts ease-of-adaptation. We then take a closer look at which linguistic attributes of a dataset predict its ID, finding that not only several shallow linguistic descriptors but also grammatical and lexical structure enable geometric compression. Finally, we find that, qualitatively, ID tends to be stable across model sizes and types , and comment on the differences between ID estimators.

### 4.1 ID tracks information-theoretic compression

Information-theoretic and geometric description length of data under OPT are Spearman-correlated. As shown for OPT-6.7b in fig. 2a, data PPL predicts ID, $\rho = 0.51$ ($p = 0.01$). The significant positive correlation between PPL and ID, moreover, holds for all model sizes tested: $\rho = 0.66$, $p < 0.01$ for OPT-350m and $\rho = 0.49$, $p = 0.01$ for OPT-1.3b, the correlation being most salient for the smallest model (see figs. E.3a and E.3d). We hypothesize that model optimization in higher dimensions permits discovery of better representa-

---

[1]We also experimented with ensembling ID metrics, but found the various ensembling methods hard to justify as NN-based estimators are very over-represented in our panel.

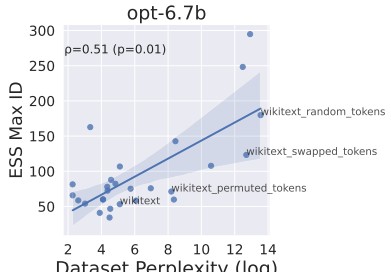 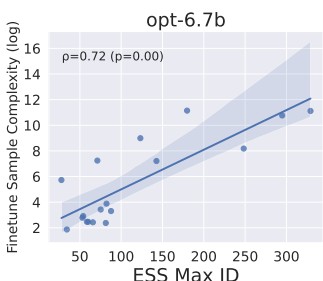 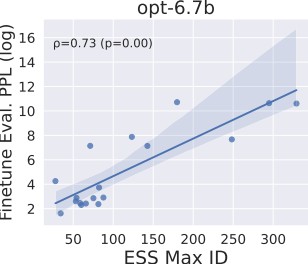

(a) Intrinsic dimension (ESS max ID) vs. data perplexity (log-PPL).

(b) Sample complexity vs. Intrinsic dimension (ESS max ID)

(c) Final log-PPL vs. Intrinsic dimension (ESS max ID).

Figure 2: Intrinsic dimension is significantly positively correlated to (a) dataset PPL ($p = 0.01$); and also predicts ease-of-adaptation metrics (b) sample complexity ($p < 0.01$) and (c) final evaluation PPL ($p < 0.01$). Both ID and dataset perplexity in (a) are measured before finetuning; in (b) and (c), ID is measured before finetuning and the y-axes after finetuning.

tions in $d$-dimensional intrinsic space, as evidenced by the correlation between performance and model size in LM scaling laws. Then, on a given dataset, larger model sizes may encounter an ID floor effect, thus weakening the correlation between PPL and ID compared to smaller sizes.

## 4.2 Compression is linked to Ease-of-Adaptation

As evidenced by the positive trends in figs. 2b and 2c, ID predicts sample complexity ($\rho = 0.72$, $p < 0.01$) as well as final PPL after convergence ($\rho = 0.73$, $p < 0.01$). These trends, moreover, are robust to model size: ID predicts sample complexity at $\rho = 0.61$, $p = 0.01$ for OPT-1.3b and $\rho = 0.81$, $p = 0.01$ for OPT-350m (figs. E.3b and E.3e); and ID predicts final PPL after finetuning at $\rho = 0.65$, $p < 0.01$ for OPT-1.3b and $\rho = 0.81$, $p = 0.01$ for OPT-350m (figs. E.3c and E.3f).

Results indicate that data which are more compressed zero-shot under the LM are easier to adapt to. Moreover, they corroborate findings in Aghajanyan et al., 2021, in which low *parameter* ID in pre-trained LMs predicts rapid adaptation. We hypothesize that this is because intrinsic data rank bottlenecks intrinsic parameter rank and vice-versa (see Rozza et al., 2012 for discussion).

## 4.3 Linguistic Correlates to Compression

In pre-trained OPT, geometric compression can be explained partially by data perplexity. But, taking a closer look, ID also correlates with interpretable linguistic descriptors such as syntactic structure or token entropy.

|        | $V$      | $\mathcal{H}_V$ | $\tilde{L}$ | $N_{tok}$ |
|--------|----------|-----------------|-------------|-----------|
| Max ID | 0.50     | 0.44            | 0.15        | 0.15      |
|        | $p$=0.01 | $p$=0.02        |             |           |

Table 2: For OPT-6.7b, Spearman correlations $\rho$ between ESS max ID and dataset vocabulary size ($V$), vocabulary entropy ($\mathcal{H}_V$), average sequence length $\tilde{L}$, and size in tokens $N_{tok}$. Significant correlations at $\alpha = 0.05$ are displayed with the corresponding $p$-value.

**Linguistic Structure Permits Compression**   Ablating linguistic structure increases ID and perplexity of data representations. This may be seen in fig. 2a for OPT-6.7b, where both the ID and perplexity of the permuted, swapped, and random variants increase with respect to the baseline for the wikitext dataset. Moreover, this relationship holds for all model sizes, see figs. E.3a and E.3d, and generally for other ID estimators, see appendix E. This indicates that learned grammatical and lexical structure permits compression in LMs; furthermore, the small increase in ID from wikitext to wikitext-permuted compared to other variants suggests that bag-of-words lexical semantics, rather than complex syntactic structure, accounts for much of geometric compression. Interestingly, this ties in with general estimates of the relative information load of syntax and lexical semantics in human language (Mollica and Piantadosi, 2019).

**Some Shallow Linguistic Descriptors Predict ID but Are Uncorrelated to Perplexity**   Beyond linguistic structure, two related shallow linguistic descriptors also predict ID: vocabulary size $V$ (number of unique tokens in the dataset) and vocabulary entropy $\mathcal{H}_V$ (Shannon entropy of token frequency

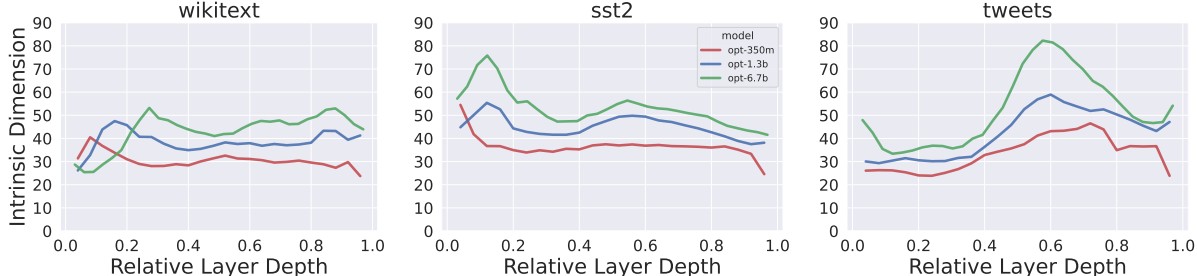

Figure 3: Example evolution of ID (ESS) over layers for three tasks (left to right): wikitext, sst2, and tweets. In general, ID profiles can be dissimilar for different datasets under the same model, and ID profiles for OPT-350m, 1.3b, and 6.7b appear correlated, with larger extrinsic dimension lending itself to slightly (not proportionately) larger intrinsic dimension.

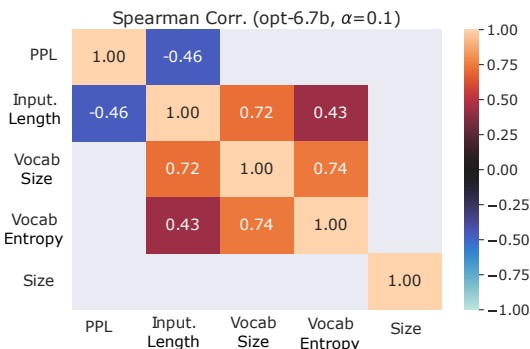

Figure 4: Spearman correlations between data descriptors: information-theoretic descriptor PPL (top/left), and shallow linguistic descriptors (bottom/right); only correlations significant at $\alpha = 0.1$ shown. Shallow metrics highly correlate to each other but not to PPL.

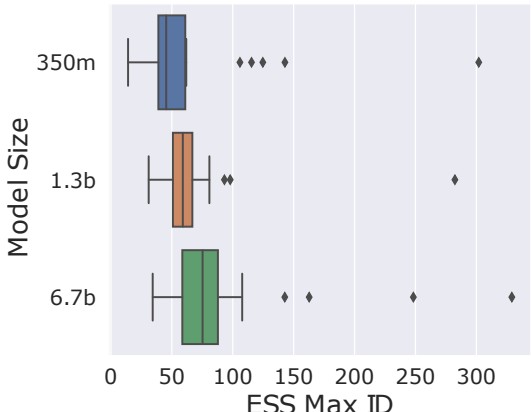

Figure 5: Distributions of max ID (aggregated over layers of the Transformer) across tasks, computed using the ESS estimator. Despite the model extrinsic dimension doubling (ED = 1024, 2048, 4096 from top to bottom), the ID remains fairly stable. Apart from a few outliers, the max ID over layers is less than 100, that is, all IDs computed over all layers are generally $\mathcal{O}(10)$.

in the dataset), see Table 2. In contrast, average sequence length (in tokens) $\tilde{L}$ and dataset size (in tokens) $N_{tok}$ do not predict ID.[2]

That descriptors such as vocabulary size and entropy predict ID is intuitive: with, e.g., a larger vocabulary, more information needs to be encoded. Furthermore, as expected, these descriptors are correlated to one another. However, the descriptors are *not positively correlated* to PPL (fig. 4), suggesting that the relationship between information-theoretic and geometric compression is not explained by shallow dataset properties.

The relation between shallow linguistic descriptors and ID generally holds across model size and ID estimators; see appendix E for further discussion, extending to other layer-aggregate measures of ID.

<hr>

[2]It is crucial that the dataset be big enough to prevent spurious correlations between size and ID due to scaling effects (see appendix A.2).

## 4.4 Intrinsic Dimension of Representations

We have presented ID as it relates to information-theoretic compression and ease-of-adaptation; now we address the question of geometric compression itself. First, we show that the various models compress data into similar ranges of ID *regardless of extrinsic hidden dimension*, lending weight to the manifold hypothesis for linguistic data. We further report on how ID estimation with different methods produces a complicated picture, and we comment on results evaluated on the full battery of 12 estimators.

**Different model size, similar ID**  We find that all tested models compress data to a similar range of ID, around $\mathcal{O}(10)$. Although the model hidden dimension doubles from OPT-350m to 1.3b to 6.7b, the range of data representational ID does not sig-

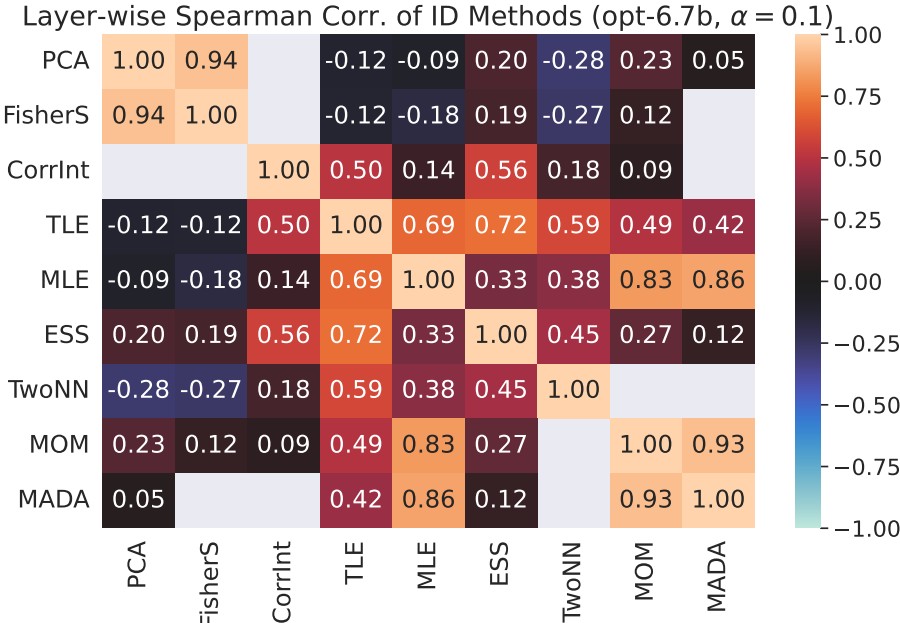

Figure 6: Spearman correlations between ID metrics: the bottom-right block of correlated metrics correspond to NN-based methods, while the top-left are PCA, Fisher Separability, and Correlation Dimension, respectively a linear projective, fine-grained clustering, and fractal method.

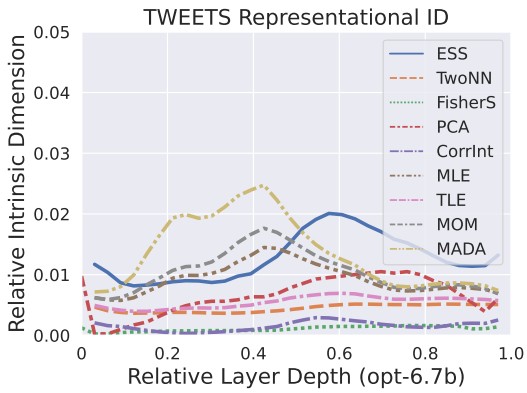

Figure 7: Different ID methods' relative ID estimates (= ID / ED) over layers in OPT-6.7b for the Tweets dataset. While some groups of methods are correlated, they produce different ID profiles for the same data.

nificantly change, see fig. 5.

Moreover, the *evolution* of ID across the layers of different model sizes follows similar trajectories: qualitatively, for all tasks, the OPT models' ID profiles follow similar global patterns (e.g., similar number of peaks), with larger models having slightly larger intrinsic dimension, see examples in fig. 3. For extended discussion of ID evolution across layers, see appendix D.

Similar ID values and evolution across different models echo results in the visual domain (Ansuini

et al., 2019) and evidence the manifold hypothesis for linguistic data. Together with past work, our results suggest that different-sized (but similar-performing) models which are trained on the same data and objective can independently recover the latent dimension of data and exhibit similar patterns of processing.

**ID estimators aren't equal; some are useful** Though ID estimators based on similar analytical methods are correlated (fig. 6), different ID estimators can produce different ID profiles for the same dataset (e.g., fig. 7). This can stem from a number of factors pertaining to assumptions and analytical method. For instance, we find the NN-based methods to be most predictive of data perplexity and ease-of-adaptation, and PCA and Fisher Separability to be least predictive. This may be because (1) PCA assumes that the underlying data manifold is linear, which may lead to poor ID estimation (consider a 1D line embedded in 2D space; PCA will estimate an ID of 2 as there are two principal directions of variance); (2) Fisher Separability systematically underestimates ID for non-uniformly distributed data (Albergante et al., 2019), and Transformer representations are highly anisotropic (Cai et al., 2021). Lastly, among the 12 estimators tested, we could not produce sensible results for three of

them (Rozza et al., 2012; Ceruti et al., 2014; Carter et al., 2010), see appendix A.3 for discussion.

While we cannot claim that any single estimator produces the "true" ID, it appears that, for purposes of ID estimation of linguistic datasets as encoded in LMs, NN-based methods are the most *useful* ones, being reliable predictors of information-theoretic compression and ease-of-adaptation (appendix E). More generally, the differing results obtained with various ID estimators reveal a need to validate them against linguistic data, which may violate underlying assumptions of common estimators, such as the global isotropy assumption in Fisher Separability (Albergante et al., 2019). While there indeed exist ID estimation benchmarks for synthetic manifolds and image data, and while NN-based estimators outperform linear ones in these benchmarks (Campadelli et al., 2015), a benchmark has not yet been developed for linguistic data, to our knowledge.

## 5 Discussion

We have quantified geometric compression in neural language models using the ID of representations, where ID tracks Shannon information-theoretic coding length. This bridges two notions of description length in pre-trained neural LMs by showing they are significantly positively correlated. Our result has also practical implications, suggesting that ID and perplexity predict how easy it is to finetune a model to a task (similarly to what observed in the context of zero-shot prompting by Gonen et al., 2022). More speculatively, the relation between ID and task adaptation may inform future modeling work that actively encourages data compression at training time, to indirectly inject fast-adaptation capabilities into a model.

ID estimators are not equal: we focus on NN-based methods because they explain useful properties of the data and ease-of-finetuning. Our work is a first attempt to evaluate a wide range of ID estimators on natural language representations, and reveals the need for a further principled study of ID estimation of linguistic data.

That nonlinear ID estimators predict information-theoretic compression and ease-of-adaptation over linear ones highlights a need to go beyond PCA in analyzing compression of specific linguistic phenomena. Our experiments on wikitext and variants also demonstrate a need for further experiments on, e.g., idiomaticity, or specific linguistic constructions (cf. Hernandez and Andreas, 2021 for analysis using PCA).

While our work investigates the relationship between ease-of-adaptation and *zero-shot* compression, a logical next step is to investigate how finetuning dynamically affects data compression under the model. We hypothesize that the result may depend on whether the dataset used for finetuning is memorized by the model during training.

Finally, while LMs are trained to reproduce the distribution of human language, it is yet unclear whether analyzing the linguistic representations of the former allows us to make statements about the dimensionality of the latter. Then, an open question remains: what is the "true" dimensionality of natural language, and to what extent do LMs recover it?

## Limitations

- While we confirmed that modern LMs do compress language data, and that this compression is correlated with ease-of-learning, we only provided a limited characterization of the relation between LM compression and linguistic properties of the input, such as lexical information and syntactic structure.

- Due to both access restriction and computational limitations, we cannot replicate our investigations on huge language models such as ChatGPT.

- A related question is to what extent the correlation we report between compression and ease of learning would hold (or even how it could be meaningfully formulated) in the context of prompt-based zero-shot task adaptation as afforded by huge LMs.

- More generally, it remains to be explored how a number of modeling choices, such as non-causal predictive objectives or instruction tuning, would affect our generalizations.

## Ethics Statement

This paper does not introduce new models or datasets, and it presents an abstract analysis of language model data compression that, we think, should not raise ethical concerns. We believe on the other hand that our focus on improving the understanding of how language models process information can be generally beneficial, in an AI landscape in which powerful language models are deployed with little understanding of the mechanics by which they work, and, consequently, little ability to control their behavior.

## Acknowledgements

We thank Alessandro Laio for generous guidance on intrinsic dimensionality estimation. Jacob Andreas provided very helpful feedback on the project. We also thank the members of the UPF COLT lab, especially Gemma Boleda, Roberto Dessì and Lucas Weber for early feedback, the members of the Barcelona Apple Machine Learning Research group and the participants in the EviL seminar for helpful feedback and suggestions. Our work was funded by the European Research Council (ERC) under the European Union's Horizon 2020 research and innovation programme (grant agreement No. 101019291). This paper reflects the authors' view only, and the ERC is not responsible for any use that may be made of the information it contains.

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

# A ID Estimation

In this appendix, we detail the tested ID estimation methods (appendix A.1) and validate their convergence on a sample of $50,000$ datapoints of the bookcorpus dataset in appendix A.2.

## A.1 ID Estimator Details

For each ID estimator tested, we provide a brief description as well as the hyperparameters used (which we will denote $k$) in relation to the skdim package. As ID estimation theory is not our focus, we refer the reader to Campadelli et al., 2015 for a comprehensive review of methods.

Note that ID estimators can be classified as *global* or *local*. Global ID estimation assumes the entire dataset lies on a single geometric object, and estimates its dimensionality from the entire dataset at once. In contrast, local ID estimation computes point-wise ID of data neighborhoods, then aggregates the per-neighborhood estimates to arrive at a global estimate. This aggregation of local estimates can correct for negative biases of global ID estimators in high $d$, where this bias is symptomatic of concentration of measure in high dimensions (Carter et al., 2010). While Mamou et al., 2020 show that linguistic representations lie on separable manifolds, we assumed for simplicity that our data lie on one geometric object and use either global or local methods to estimate its ID, leaving per-manifold ID estimation to future work.

| Estimator | Method | Global/Local |
|---|---|---|
| PCA | Projective | Global |
| FisherS | Fine-grained clustering | Global |
| CorrInt | Fractal | Global |
| TwoNN | NN-based | Global |
| KNN | NN-based | Global |
| MiND ML | NN-based | Global |
| DANCo | NN-based | Global |
| TLE | NN-based | Local |
| MLE | NN-based | Local |
| MOM | NN-based | Local |
| MADA | NN-based | Local |
| ESS | NN-based | Local |

Table A.1: Taxonomy of 12 ID estimators tested (3 of which unsuccessfully: KNN, MiND ML, and DANCo), by method and by whether global or local. If local, point-wise ID estimates are computed and mean-aggregated.

### A.1.1 Global Estimators

**PCA** Projective method that assumes a linear manifold. The ID $d$ is the number of principal values $\lambda > \frac{\lambda_{max}}{k}$, where hyperparameter $k = 20$ (Fukunaga and Olsen, 1971).

**Fisher Separability (FisherS)** Fine-grained clustering estimator that relies on "clumping" phenomena in high dimensions. To estimate $d$, (1) the data are transformed using PCA (hyperparameter $k = 10$) and projected onto the unit sphere; (2) the proportion of linearly separable datapoints is compared to that of a theoretical equidistribution on a $d$-sphere (Albergante et al., 2019).

**Correlation Dimension (CorrInt)** The volume $s$ of a $d$-dimensional radius-$r$ hypersphere grows as $s \sim r^d$. For two values of $r$: $r_1$ and $r_2$, estimate $s$ by counting the nearest neighbors in a radius-$r$ sphere around each point. Then, fit $d$ (Grassberger and Procaccia, 1983).

We choose $r_1$ corresponding to hyperparameter $k_1 = 10$ nearest neighbors and $r_2$ corresponding to hyperparameter $k_2 = 20$ nearest neighbors.

**TwoNN** A nearest-neighbors (NN)-based estimator that assumes local uniform data density (up to the second neighbor). Regresses $\frac{\log(1-F(\mu))}{\log \mu} = d$, where $F$ is the cumulative density function and $\mu_i := \frac{r_2}{r_1}$ is the 2NNs distance ratio for each $x_i \in X$. Following Facco et al., 2017, we discard hyperparameter $k = 0.1$ fraction of largest $\mu_i$ from the regression, a heuristic that improves estimation accuracy.

### A.1.2 Local Estimators

**ESS** For each datapoint, take hyperparameter $k = 10$ nearest neighbors. This neighborhood is assumed to be approximable by a uniform distribution of datapoints on a tangent plane to the manifold. Expected Simplex Skewness, or ESS (called *ESSa* in Johnsson et al., 2015), constructs a simplex with one vertex at the centroid of the local dataset and other vertices at the other datapoints. Then, the expected simplex skewness measure is defined as the volume of the simplex divided by the volume if all edges to the centroid were orthogonal. The ESS is computed over all such local datasets then compared to the theoretical expected value in $d$-dimensions in order to estimate $d$.

**TLE** Tight Local ID Estimation consists of computing a modified Correlation Dimension on small

neighborhoods of hyperparameter $k = 20$ neighbors around query datapoints, then fitting $d$ using maximum-likelihood estimation (Amsaleg et al., 2019).

**MLE**  We use the Maximum Likelihood Estimator of Haro et al., 2008, which extends Levina and Bickel, 2004 to handle noise. For simplicity, we assume one underlying data manifold (the method extends to multiple underlying manifolds). Then, MLE models the number of points falling into a small neighborhood around query points by a (Translated) Poisson process. The maximum-likelihood estimator for the Poisson $\lambda$ parameter is fit given data, from which one can solve for the local ID at that neighborhood. Finally, local IDs are averaged to arrive at the overall estimate.

**MOM**  The Method-of-Moments estimator from Amsaleg et al., 2018 takes as reference the distribution of distances $X$ to a query point $q$, over hyperparameter $k = 100$ NNs of $q$. The procedure estimates empirical moments of $X$ and compares them to the theoretical value, which is a function of $d$, in order to solve for $d$.

**MADA**  Manifold Adaptive Dimension Estimation (MADA) compares the empirical probability that datapoints fall into a ball around a query datapoint to the theoretical probability, which is a function of $d$. More precisely,

$$\mathbb{P}(x_i \in B(x, r)) = \eta(x, r)r^d$$

for some function $\eta$, where for small-enough $r$, $\eta(x, r)$ is essentially constant. Then, $d$ is fit over many query points $x$ and $x_i$ in the dataset (Farahmand et al., 2007).

## A.2  ID Estimation Convergence Analysis

In section 3.1, we state that ID estimation is performed for datasets $X$ of size $N > 10000$. For large datasets ($N > 50000$), we compute ID on random subsample of $\min(N, 50000)$ data points for efficiency. Here, we verify that ID computed on a sample of size $10000 < N < 50000$ for $\max(D) = 4096$ reasonably converges, the reason being that NN-based ID estimators can be *scale-dependent*. Scale-dependence, or sensitivity to data resolution, of NN-based estimators is well-documented in the literature. Due to concentration of measure in high dimensions, these estimators are sensitive to data density and underestimate ID for $d \gtrsim 10$ (Facco et al., 2017; Ansuini et al., 2019;

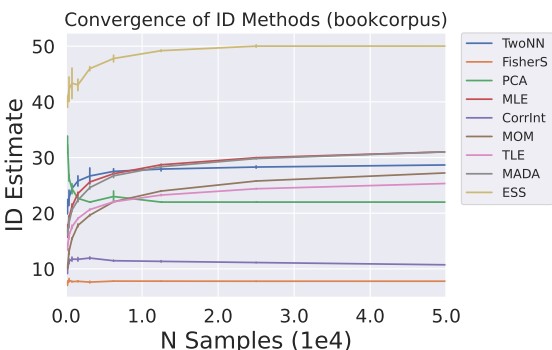

Figure A.1:  Convergence of ID estimates for bookcorpus on the last layer of OPT-6.7b, from $N' \approx 200$ to $N' = 50000$. Each datapoint is the mean ID estimate, shown with one standard deviation, on a boot-strapped sample of size $N'$, computed over 3 random seeds.

Ceruti et al., 2014), see Erba et al., 2019 for discussion.

Similar to Ansuini et al., 2019, we justify our choice of 50000 with a convergence analysis of ID estimators on a large, complex dataset, that is, bookcorpus ($N = 74004228$), using OPT-6.7b ($D = 4096$), and a one layer representation (the last). Starting at $N' = 50000$, we systematically subsample the representations on a log-scale and show that ID estimates averaged over three random seeds appear to converge for all estimators by $N' \approx 10000$ (see fig. A.1). Convergence of these estimates permits us to reliably include them in our analysis linking ID to PPL and ease-of-adaptation.

Finally, our approach is limited for small datasets, as $N$ needs to grow exponentially in $d$ in order to estimate $d$ (Bishop, 2007). As attested in the literature (Campadelli et al., 2015; Albergante et al., 2019), several ID estimators are empirically more robust to this curse of dimensionality, such as non-NN-based estimators PCA and FisherS, which converge earlier than $N' \approx 10000$. Moreover, note that for the remaining ID estimators, for small $N' \lessapprox 10000$, they are increasing, thus systematically underestimating ID, until starting to converge at around $N' \approx 10000$. Therefore, while $N' = 10000$ is not an end-all be-all cut-off for ID convergence of linguistic data, we employ it as a heuristic for dataset selection.

## A.3  Other Estimators Tested

We tested a total of 12 estimators. In addition to the 9 discussed in appendix A.1, we tested three NN-based global estimators, DANCo (Ceruti et al.,

2014), MiND ML (Rozza et al., 2012), and kNN (without bootstrapping) (Carter et al., 2010), but encountered difficulty tuning the hyperparameters to produce sensible ID estimates (MiND ML produced $d = 10$ for almost all layers on all datasets; the other estimators produced $d \approx D$). For this reason, we omit them from discussion of the results.

Finally, in future work, it is worth testing the recent FCI estimator of Erba et al., 2019 and the Generalized Ratios ID estimator of Denti et al., 2022, which are more robust to scaling effects by design.

## B  Dataset Details

In this appendix, we outline how [Super]GLUE data are preprocessed for PPL and ID computation, and describe the remaining corpora in table B.2. All datasets are freely available on HuggingFace at the time of writing.

**Preprocessing [Super]GLUE** GLUE and Super-GLUE are two linguistic classification benchmarks designed to test language models on tasks such as textual entailment and semantic equivalence (Wang et al., 2019b,a). For example, in Quora Question Pairs (qqp), two questions are given to the model as natural language, and the model is tasked to predict whether they are equivalent. If there are multiple inputs, we treat them as individual data points in the ID / PPL computations and feed them to the model separately as in Wang et al., 2019a, leaving other strategies, such as concatenation with delimiters (Brown et al., 2020), to future work.

## C  Finetuning Details

We implement full finetuning for OPT-350m and parameter-efficient finetuning using LoRA (Hu et al., 2022) for the other LMs on a standard cross-entropy causal language modeling objective, optimized using AdamW (Loshchilov and Hutter, 2019).

To avoid an expensive hyperparameter search, we set a constant batch size for each model (the largest fitting in memory), with a constant learning rate schedule. Then, the only hyperparameter we vary is the learning rate from 5e-5, 5e-6 to 5e-7. For each learning rate, we average final evaluation PPL after training over three random seeds, and we choose the best learning rate by the lowest final evaluation PPL. All values reported are averaged over 3 random seeds for the best learning rate.

| Corpus | Summary |
|---|---|
| IMDB (Maas et al., 2011) | Movie reviews |
| Penn Treebank (Marcus et al., 1993) | Text-only version of Penn Treebank (content from 1989 Wall Street Journal) |
| Bookcorpus (Zhu et al., 2015) | Text from books |
| Wikitext (Merity et al., 2017) | Cleaned Wikipedia |
| Wikitext fr (Simoulin and Crabbé, 2021) | Cleaned French Wikipedia |
| Tweets (Barbieri et al., 2020) | Twitter dataset's sentiment evaluation subset |
| Pile-10k (Nanda, 2022) | First 10k sequences of The Pile (Gao et al., 2020) |
| Openwebtext-10k (Bekman, 2022) | 10k sequences from Openwebtext (Gokaslan et al., 2019) |
| CNN Dailymail (Hermann et al., 2015) | News articles from CNN |
| CONCODE (Soliman, 2022) | Set of Java classes from CONCODE (Iyer et al., 2018) |

Table B.2: Description of external non-[Super]GLUE corpora

| Hyperparameter | Value |
|---|---|
| Batch size | 8 (opt-350m); 16 (opt-6.7b); 32 (opt-1.3b) |
| lr | 5e-5, 5e-6, 5e-7 |
| max train epochs | 15 |
| LoRA rank | 8 |
| LoRA $\alpha$ | 8 |
| AdamW $\beta_1, \beta_2$ | (0.9, 0.999) |
| AdamW $\epsilon$ | 1e-6 |

Table C.3: Hyperparameters used in finetuning, where OPT-350m is fully finetuned and all other models are finetuned using LoRA.

Hyperparameters can be found in table C.3. Code is adapted from HuggingFace implementations and can be found at https://github.com/

## D   ID over layers

In Section 4, we have presented results focusing on max ID as our aggregate measure across layers. However, the evolution of ID across layers is an interesting problem in itself, and reveals how Transformers sequentially compress (or decompress) linguistic representations (cf. Valeriani et al., 2023 and Ansuini et al., 2019 for equivalent studies on protein sequence and vision models, respectively).

In the large majority of cases, ID is relatively stable across layers, justifying the choice of focusing on an aggregate measure: see representative examples in fig. D.2a. We do, however, also observe small clusters of datasets where the ID profile across layers exhibits different patterns. For example, for a number of datasets (example in fig. D.2b), we observe a first increase in ID in the early layers followed by a later increase at the middle layers, which might suggest kernel-function-like dimensionality expansions before new reductions, along the lines of what was observed by Ansuini et al., 2019. Another interesting pattern pertains to the OPTCorpus, showing a late increase in dimensionality. Intriguingly, this pattern is not disrupted by the ablations, as shown in fig. D.2c. Still, we want to emphasize that in most cases ID profiles are relatively flat: even when they aren't (as in figs. D.2b and D.2c), the fluctuations stay within relatively narrow ranges. We conjecture that residual connections have a stabilizing effect on ID across layers. Notably, however, Valeriani et al., 2023 found that Transformers applied to visual and protein sequence data displayed an early peak in ID that is missing in our linguistic datasets, whose profiles vary in shape. We leave a fuller understanding of the differences to future work.

## E   Extended Results

Here, we review results on the relationship between geometric compression, information-theoretic compression, and ease-of-adaptation for all model sizes, ID estimators, and aggregations of ID.

**Alternative Aggregations of ID**   In addition to the max ID over layers, which is discussed in the main article, we test the following other aggregations: min, first (ID of positional embeddings), last (ID before next-word prediction), mean, and median, which are "point-aggregates" of ID, and change (last−first) and range (max−min), which summarize the spread of ID, i.e., how data is processed in terms of expansion or reduction over layers.

We find that when one aggregation of ID is correlated to PPL, so are other aggregations. For instance, as the max ID influences the mean ID, they are often both correlated to PPL. For estimators CorrInt, TwoNN, ESS, TLE, and MLE, we notice virtually all aggregations of ID correlating to perplexity in OPT-350m (right column of figs. E.6 and E.7).

**ID vs. Perplexity**   We observe the following about the Spearman correlation between aggregations of ID and dataset perplexity:

- **Correlation strength tends to increase as model size decreases**. This is seen by looking at the leftmost columns of the correlation plots: there are more significant correlations (colored squares) as model size decreases in figs. E.6 and E.7.

- When significant, **min, max, first, last, mean, and median ID are *positively correlated* to PPL** for all model sizes and ID estimators.

- When significant, **change in ID is *negatively correlated* with PPL** for all model sizes and ID estimators. That is, intuitively, if an input is harder for the model to process (larger PPL), it projects its features into higher-dimensional space for next-token prediction, which we interpret to be a kernel-expansion-like operation à la Ansuini et al., 2019.

- When significant, **range of ID positively correlates to PPL** for all model sizes and ID estimators. This correlation can be partially explained by those between the min and max ID and PPL.

**ID vs. Ease-of-Adaptation**   We find that several ID aggregates are good predictors of ease-of-adaptation across model size and ID estimator. We observe the following:

- For Correlation Dimension and NN-based ID estimators, when significant, **point estimates of ID (e.g., max, mean) positively correlate to final evaluation PPL and sample complexity**, as evidenced by the orange-colored squares in the bottom-right of plots in figs. E.6 and E.7. This trend breaks down for PCA and FisherS, but appears robust to model size.

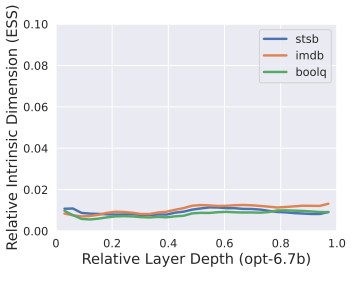 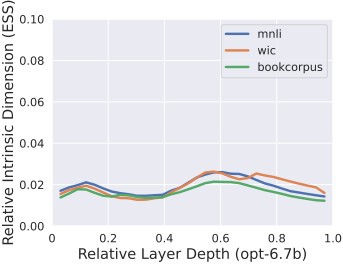 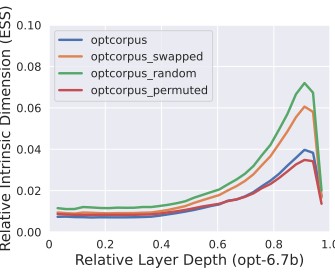

(a) Relatively stable ID      (b) Expansions then reductions in ID      (c) ID profiles for OPTCorpora

Figure D.2: Representative (relative) ID profiles, computed with ESS, of dataset representations against layer depth for OPT-6.7b, qualitatively grouped into three categories: (a) "flat" (the most common case), (b) "hilly", and (c) gradual increase then sudden decrease, corresponding to OPTCorpora. ID profiles are calculated on $\min(N, 50000)$ sequences for each dataset shown, where $N$ is the number of sequences in the dataset.

- **Results on ID spread (change and range) are inconclusive**, giving sometimes negative and sometimes positive correlations depending on the estimator and model size.

**ID vs. Shallow Linguistic Descriptors** The relationship between ID and shallow linguistic descriptors like vocab size, vocab entropy, average sequence length, and number of tokens appears to diverge by ID method category. We observe the following (see figs. E.6 and E.7):

- Trends appear to be opposite for PCA/FisherS and the other ID estimators.

- For PCA/FisherS, ID generally correlates negatively to linguistic descriptors like average input length, vocab size, and vocab entropy.

- When significant, vocab size and entropy generally correlate positively to ID aggregates for estimators that aren't PCA/FisherS.

**ID and Linguistic Structure** Beyond ESS (section 4.3), the increasing trend in ID/PPL for variants of wikitext, successively baseline < permuted < swapped $\lesssim$ random, also holds for virtually all ID estimators and model sizes (fig. E.5)– all but PCA/FisherS (not pictured in the figure), who estimate ID of all variants to be baseline = permuted = swapped < random.

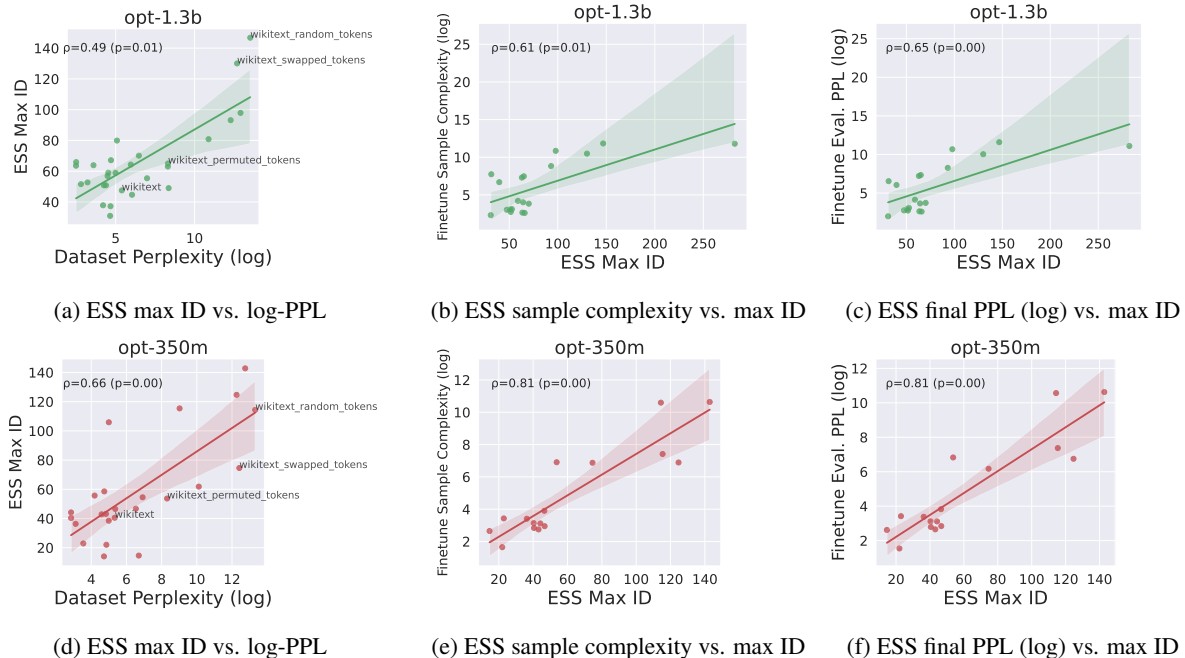

(a) ESS max ID vs. log-PPL    (b) ESS sample complexity vs. max ID    (c) ESS final PPL (log) vs. max ID

(d) ESS max ID vs. log-PPL    (e) ESS sample complexity vs. max ID    (f) ESS final PPL (log) vs. max ID

Figure E.3: For OPT-1.3b (top row) and OPT-350m (bottom row), max ID estimated with ESS is positively correlated to (a,d) dataset PPL (significant with $p \leq 0.01$ for both models); predicts ease-of-adaptation metrics (b,e) sample complexity (significant with $p \leq 0.01$ for both models); and (c,f) final evaluation PPL (significant with $p \leq 0.01$ for both models). Datapoints for wikitext and its perturbations are labeled for reference. The right two plots contain the subset of datasets tested that are suitable for a causal language modeling objective.

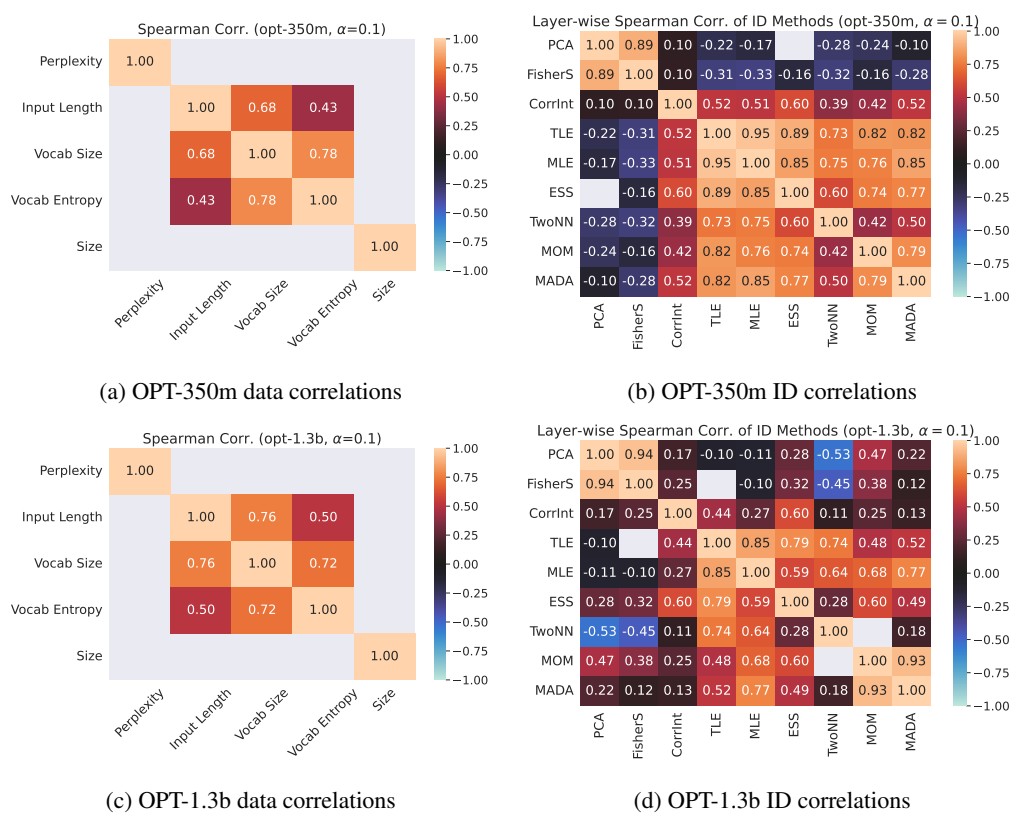

(a) OPT-350m data correlations      (b) OPT-350m ID correlations

(c) OPT-1.3b data correlations      (d) OPT-1.3b ID correlations

Figure E.4: Data metrics Spearman correlations for left column (a) OPT-350m and (c) OPT-1.3b; and ID metrics Spearman correlations for right column (b) OPT-350m and (d) OPT-1.3b. Note the inter-correlations between shallow linguistic descriptors input length, vocab size, and vocab entropy for all model sizes, as well as the grouping of ID metrics into one block corresponding to NN-based methods (bottom right of each plot).

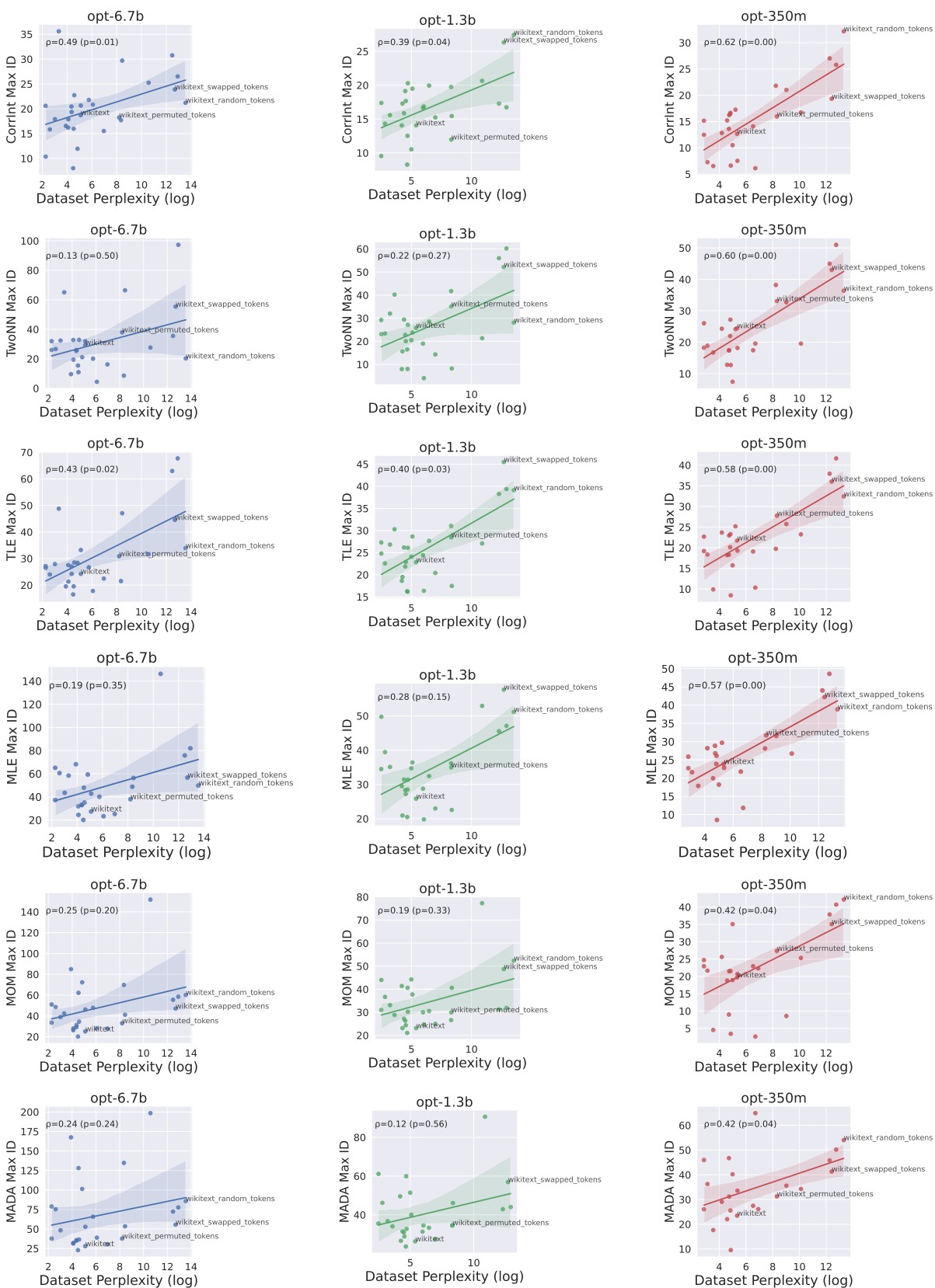

Figure E.5: For OPT-6.7b (left), OPT-1.3b (middle) and OPT-350m (right), and for all other ID estimators considered, we plot max ID vs. dataset PPL (log), labeling the points for wikitext. Across estimators and model sizes, we see a general trend that ablating linguistic structure increases ID as well as PPL, in the order of baseline $<$ permuted $<$ swapped $\lesssim$ random.

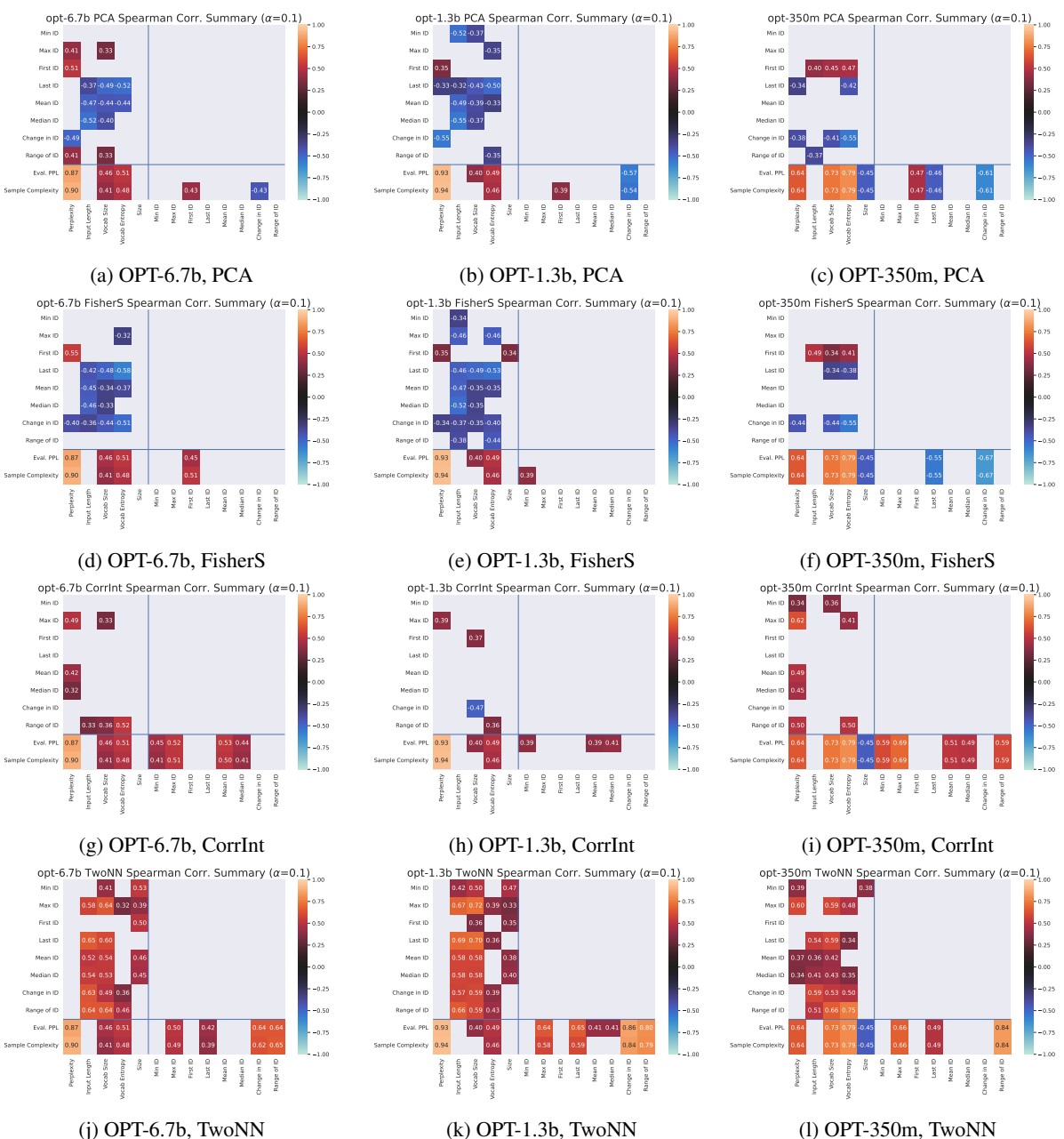

(a) OPT-6.7b, PCA      (b) OPT-1.3b, PCA      (c) OPT-350m, PCA

(d) OPT-6.7b, FisherS      (e) OPT-1.3b, FisherS      (f) OPT-350m, FisherS

(g) OPT-6.7b, CorrInt      (h) OPT-1.3b, CorrInt      (i) OPT-350m, CorrInt

(j) OPT-6.7b, TwoNN      (k) OPT-1.3b, TwoNN      (l) OPT-350m, TwoNN

Figure E.6: Global ID Estimators: full panel of Spearman correlations for models (left to right columns) OPT-6.7b, 1.3b, and 350m. Each figure is a summary of Spearman correlations given a model and global ID metric, significant at $\alpha = 0.1$, between aggregated ID and data metrics (top left), ease-of-finetuning metrics and data metrics (bottom left), and ease-of-finetuning and ID (bottom right).

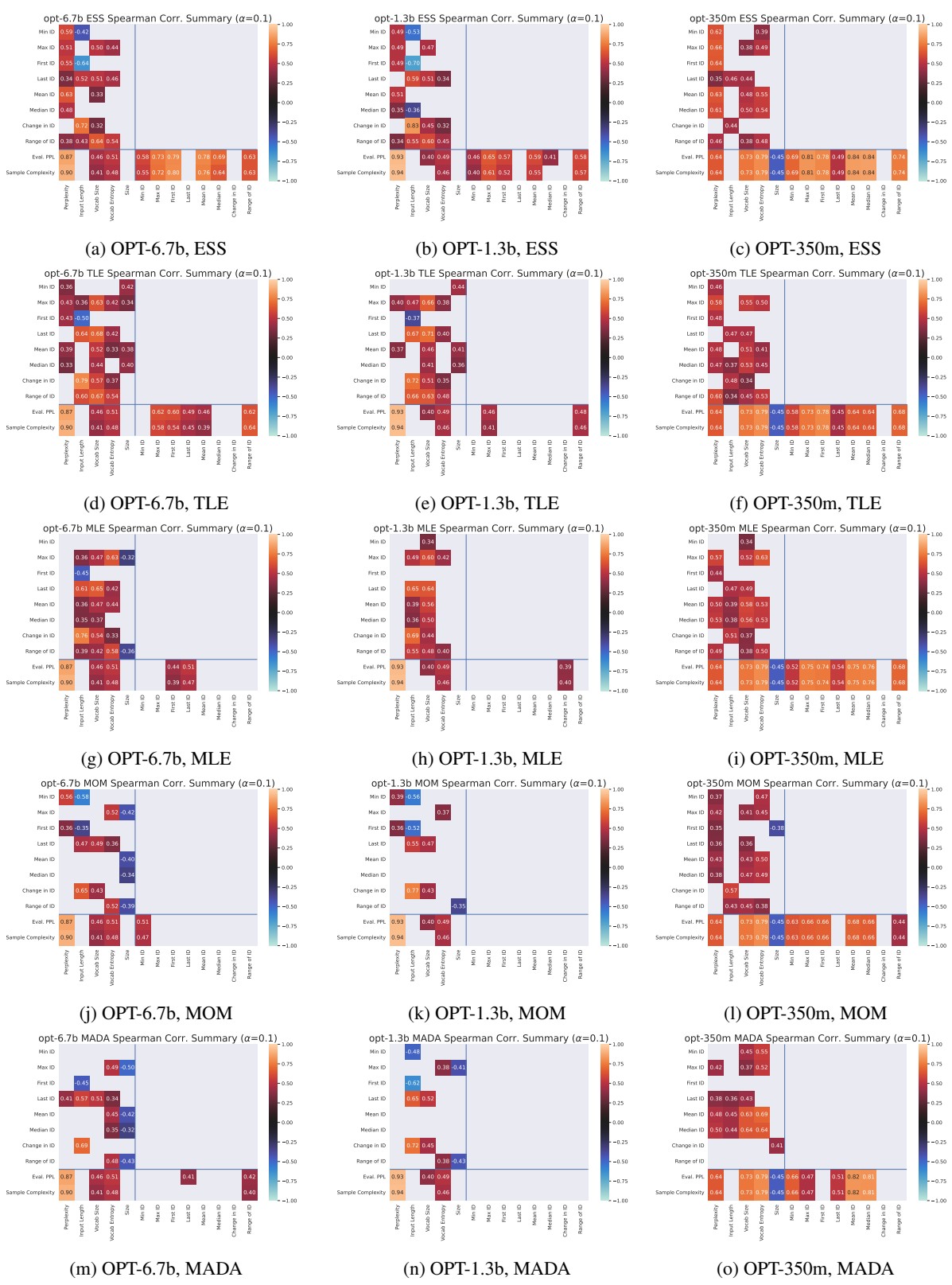

Figure E.7: Local ID Estimators: full panel of Spearman correlations for models (left to right columns) OPT-6.7b, 1.3b, and 350m. Each figure is a summary of Spearman correlations given a model and ID metric, significant at $\alpha = 0.1$, between aggregated ID and data metrics (top left), ease-of-finetuning metrics and data metrics (bottom left), and ease-of-finetuning and ID (bottom right).