# OpenReview forum: "Bridging Information-Theoretic and Geometric Compression in Language Models"
_EMNLP/2023/Conference — EMNLP 2023 Main_

### Official Review · Reviewer_LFBq · 2023-08-04

**Soundness:** 4

**Excitement:**

4: Strong: This paper deepens the understanding of some phenomenon or lowers the barriers to an existing research direction.

**Paper Topic And Main Contributions:**

This paper analyses the Language Models in terms of geometric and information-theoretic compression. In particular, it measures the correlation between geometric compression, information-theoretic compression, and ease of adaptation. For each of these properties, it proposes to use different methods, i.e., intrinsic dimension for geometric compression, perplexity for information-theoretic compression, and, sample efficiency and validation perplexity as a measure of ease of adaptation while fine-tuning. It shows that there is a positive correlation between intrinsic dimension and perplexity as well as ease-of-adaptation. Moreover, the intrinsic dimension among different LMs in size suggests that data resides in a much smaller manifold than the extrinsic dimension.

**Reasons To Accept:**

- The paper is clearly written
- The findings of the paper could be of interest to the community

**Reasons To Reject:**

- The authors could have evaluated on a wider range of LMs in order to make their arguments stronger.

**Reproducibility:**

4: Could mostly reproduce the results, but there may be some variation because of sample variance or minor variations in their interpretation of the protocol or method.

**Reviewer Confidence:**

3: Pretty sure, but there's a chance I missed something. Although I have a good feel for this area in general, I did not carefully check the paper's details, e.g., the math, experimental design, or novelty.

---

> ### Author Rebuttal · Authors · 2023-08-28
>
> Thank you for your time reading our work and for your positive response! Following your feedback, we intend to include a wider range of LMs in the camera-ready including BLOOM-560m and LLAMA 2, and experiments computing ID and PPL for BLOOM-560m are currently running. So far, the ID and PPL numbers are looking similar to those of OPT-6.7b.

---

### Official Review · Reviewer_xjpS · 2023-08-12

**Soundness:** 4

**Excitement:**

4: Strong: This paper deepens the understanding of some phenomenon or lowers the barriers to an existing research direction.

**Paper Topic And Main Contributions:**

The authors investigate the relationship between compression on data representations and network reduction.
They find that correlation between adaption of casual language models and data compression. They also show an ablation study of different techniques and the extent to which they capture the relationship beween the intrisic dimension of the data, coding length and ease of adaptation for casual models.

**Reasons To Accept:**

Their ablation study of the relationship between compression on data representations and network reduction.

**Reasons To Reject:**

None

**Reproducibility:**

4: Could mostly reproduce the results, but there may be some variation because of sample variance or minor variations in their interpretation of the protocol or method.

**Reviewer Confidence:**

4: Quite sure. I tried to check the important points carefully. It's unlikely, though conceivable, that I missed something that should affect my ratings.

---

> ### Author Rebuttal · Authors · 2023-08-28
>
> Thank you for your time reading our work! We appreciate that you found the ablation study interesting.

---

### Official Review · Reviewer_78nc · 2023-08-12

**Soundness:** 4

**Excitement:**

4: Strong: This paper deepens the understanding of some phenomenon or lowers the barriers to an existing research direction.

**Paper Topic And Main Contributions:**

This paper analyzes the relationship between geometric compression (as indicated by the intrinsic dimensionality), information-theoretic complexity (measured as perplexity) and ease of adaptation (measured as perplexity after fine-tuning) for the representations learned by large language model. Their experiments show that geometric compression predicts information-theoretic complexity and ease of adaptation. They verified that this relationship holds even when linguistic structure is progressively destroyed -- ablating syntax and semantics increases both perplexity and intrinsic dimensionality. Furthermore, their results also indicate that despite having different extrinsic dimensionality (defined by the representation size), LLMs tend to have similar and rather small intrinsic dimensionality that is in O(10).

**Questions For The Authors:**

A. It is not clear to me that the datasources are diverse enough. Most datasets seem to contain factual information and are likely to contain simple sentences. How likely is it that the results would differ on datasets containing a higher proportion of complex sentences, like classic novels, etc.?

B. To what extent are the effects of ablating syntax related to the choice of representation? In your opinion, would using a different representation, like an average or concatenation of all the token embeddings change any trends? I know this is out of scope of this paper but informed speculation would still be helpful.


**Reasons To Accept:**

- Experiments are thorough and expansive, with variety of estimation methods, and datasets being tested, which lends credence to results.
- Although many of the results presented in the paper may have been expected, it is valuable that they have been demonstrated and quantified through rigorous experimentation.
- This paper might stimulate interest in analyzing linguistic characteristics via LLMs that might also enhance our understanding of human language.

**Reasons To Reject:**

1. Lack of diversity in models. All the models are of the OPT family. Perhaps models with different training data like BLOOM, or different architectures and training methodolgies like Llama should have been included to marginalize out the impact of modelling choices made in OPT.
1. Lack of clarity in some places:
    1. It is unclear if the model is fine-tuned on the transformed dataset.
    1. It is unclear which results are based on the fine-tuned models. Does Fig 2 (a) use fine-tuned models, or does it simply evaluate the pretrained model with these datasets?
    1. Zero-shot perplexity is not defined.

**Reproducibility:**

4: Could mostly reproduce the results, but there may be some variation because of sample variance or minor variations in their interpretation of the protocol or method.

**Reviewer Confidence:**

3: Pretty sure, but there's a chance I missed something. Although I have a good feel for this area in general, I did not carefully check the paper's details, e.g., the math, experimental design, or novelty.

---

> ### Author Rebuttal · Authors · 2023-08-28
>
> Thank you for your constructive feedback! We are glad you found our work to be interesting and thorough. To respond to your points,
>
> 1. In light of limited computational resources, we chose to focus on experiments spanning a breadth of model sizes and datasets. We agree, though, that our results can be strengthened by testing different model families. Therefore, towards including a more expansive set of experiments in the camera-ready, we have thus far launched experiments for BLOOM-560m this week, so far finding similar ranges of perplexity and ID to OPT-6.7b.
> We plan to replicate the full experiments for BLOOM-560m and for LLAMA 2 to include in the appendix of the camera-ready.
>
> 2. __Clarity points__ (we will also clarify them in the revision):
>
>     a. We state in l338-341 that, like all other datasets that are not [Super]GLUE, the models are fine-tuned on the transformed datasets (e.g. wikitext-permuted) so as to compute their finetuning sample complexity and final PPL.
>
>     b. Fig 2a uses models before finetuning, or the "zero-shot perplexity" mentioned in your next comment. We will make this clear in the figure caption.
>
>     c. We agree that (l359) zero-shot perplexity may be unclear. We'll change it to "zero-shot perplexity ($PPL_0$), or perplexity evaluated on the pre-trained model before finetuning".
>
> 3. __Questions__
>
>     A. We took care to include a variety of data sources, including one containing classic novels (bookcorpus, see Table 1), as well as those which are "non-language" (the syntax-ablated datasets), a Java code dataset, encyclopedic data, encyclopedic data in a different language (wikitext french), data scraped from the web (openwebtext-2), etc, which we believe represent a diverse set of language that covers in-distribution and out-of-distribution for the model. Moreover, the data are diverse in terms of perplexity and ID, which is the main focus of our study.
>
>     On a more general note, we think that having, e.g., a high proportion of complex sentence structure or rare vocabulary items is interesting insofar as it is a __distribution shift__ from the model's learned distribution of language. In future work, we definitely plan to experiment with these distribution shifts in a controlled way, seeing how types of shifts, e.g. increasing syntactic complexity or changing word frequency, affect perplexity and ID.
>
>     B. Thanks for the interesting question! We would like to try other representations that take all tokens into account. It is possible that the choice of representation will change the sensitivity of the analysis to syntax ablation. Note that averaging all of the token embeddings might actually make the representation less sensitive to syntactic structure, as it muddles information about word ordering. A concatenation-based representation is more promising, but we need to solve the technical issue of how to compare sentences/paragraphs with a different number of tokens in this setup.

---

### Meta-Review · Area_Chair_LbVn · 2023-09-20

**Recommendation:** 5

**Metareview:**

The reviewers were unanimous that this is a solid paper, and I concur. To add to their notes, I appreciate the distinctive hypothesis that this tests relative to much of the analysis work in NLP.

As for positives, reviewers note the rigor and likelihood of stimulating discussion and understanding of LMs. For negatives, reviewers wish for more LMs in the evaluation; I’m not overly concerned about this. I think the paper is rigorous and relatively clear, but additional work could be put into presentation of the various compression metrics and their evaluation (some nice figures would help.)

Adding my own concern, I think the destruction of linguistic structure through vocabulary swapping etc. is rather severe, and so the increase in intrinsic dimensionality seems less interesting. I encourage the authors to consider some less severe transforms, like noisy paraphrase.

The limitations section notes the disconnect between the somewhat lofty discussion of infinite use of finite means in the introduction and the concrete experiments in the paper; I think that walking readers along the connections would help this!

---

### Decision · Program_Chairs · 2023-10-07

**Decision:**

Accept-Main

**Comment:**

The reviewers were unanimous that this is a solid paper, and I concur. To add to their notes, I appreciate the distinctive hypothesis that this tests relative to much of the analysis work in NLP.

As for positives, reviewers note the rigor and likelihood of stimulating discussion and understanding of LMs. For negatives, reviewers wish for more LMs in the evaluation; I’m not overly concerned about this. I think the paper is rigorous and relatively clear, but additional work could be put into presentation of the various compression metrics and their evaluation (some nice figures would help.)

Adding my own concern, I think the destruction of linguistic structure through vocabulary swapping etc. is rather severe, and so the increase in intrinsic dimensionality seems less interesting. I encourage the authors to consider some less severe transforms, like noisy paraphrase.

The limitations section notes the disconnect between the somewhat lofty discussion of infinite use of finite means in the introduction and the concrete experiments in the paper; I think that walking readers along the connections would help this!